# Victimizations and surviving of workplace violence against waitresses in southern Ethiopia

**Bewunetu Zewude** [ORCID]*, **Tewodros Habtegiorgis**

Department of Sociology, Wolaita Sodo University, Sodo, Ethiopia

* bewunetuzewude@gmail.com

## Abstract

Women are obliged to take on vulnerable forms of employment that fail to protect their basic labor rights. Exposure to workplace violence is especially high for those women who work within the agriculture, the hotels, restaurants and catering, the transport as well as the manufacturing sectors. In this context, we explored workplace violence against women working as waitresses in various hotels, restaurants, cafeterias and grocery stores of Wolaita Sodo town, southern Ethiopia. Cross-sectional study was undertaken with descriptive and exploratory study designs. Qualitative data were collected by using in-depth interview method in which 16 waitresses were interviewed. Data were voice recorded, transcribed, and analyzed searching themes and patterns in the data. While we found that waitresses are highly exposed to workplace violence, the level of exposure to the violence varies across various circumstances, including working in large and small towns, the situation of the owners/supervisors, public's insight of the position, waitresses' ability to speak the local language, the role of the waitresses, customers' behavior and the specific context in which waitresses work. Waitresses were generally exposed to all forms of violence including spitting, throwing objects, simple physical assault, touching on private parts, and intimidation, among which verbal abuse and emotional exploitations are found to be the most common. Furthermore, the results revealed that though waitresses rarely experienced violence from female customers, the most common perpetrators involved in the practice were males. Amid widespread exposure of waitresses to all forms of workplace violence, ignorance, mostly motivated by fear of losing one's job, has been the common way waitresses respond to the violence. The study implies the vulnerability state of waitresses partly due to lack of adequate awareness regarding the rights and obligations expected of an employee when working in such organizations. This is again exacerbated by the lack of formal employment procedures in such organizations. Therefore, awareness creation and supervisory activities are expected from the media, workers' and social security office of the government, police and other concerned bodies.

**Data Availability Statement:** All relevant data are within the manuscript and its Supporting information files.

**Funding:** The authors received no specific funding for this work.

**Competing interests:** The authors have declared that no competing interests exist.

# Introduction

As a result of their increasing participation in the economic activities, women's reliance on their husbands has decreased and their socio-economic status is improving through time [1]. Women are increasingly participating in the productive economic activities, including the formal sector, representing more than 42% of the global paid working population [2,3]. Following their increased involvement in the public sphere, women meet new health hazards [4]. Women's participation in the labor force shows the importance and contribution of women to economic productivity, hence, the need for occupational health and safety concerns [5]. However, the beneficial effect of women's participation in economic activities on the lives partly depends on the nature of work and the work environment, among other things [6].

Though all workers are exposed to occupational violence, women are greatly affected and relatively more vulnerable to workplace violence compared to their men counterparts [7]. Gender inequalities in the workplace have become increasingly important following more women joining the work force [8]. Although men tend to be at greater risk of direct physical assault as a result of their engagement in the commonly 'men's jobs,' such as women are found in many of the occupations with a high-risk of sexual harassment, violence and threats of violence, working in contact with the public in banks, bookmakers, shops and in solitary settings [5,9,10]. According to [11], violence against women is a means of control and oppression by men. Above all, workplace violence should be understood as a reflection of the general phenomenon of violence in many areas of social life [12].

Gender-based violence against women is the most prevalent form of abuse worldwide, affecting one third of all women in their lifetime [13]. It ranges from sexual harassment to rape, domestic violence to trafficking [14]. It is recognized as a health, economic development, and human rights concern that both reflects and reinforces inequalities between women and men [15,16]. A study undertaken in Zimbabwe, for instance, uncovered the impact of gender-based violence on the reproductive health of women [17]. In addition, the negative impacts of gender discrimination on development and poverty alleviation has been found by [18]. While factors such as working alone, working with the public, and in educational settings have been identified as task-related risk factors, the working environment such as organizational setting and managerial style, as well as the workplace culture are workplace risk factors for the prevalence of workplace violence against women [16]. [19] maintain that working with customers or the public is among the factors that put people at increased risk of workplace violence and sellers of alcohol are among the occupational groups that tend to be at risk from workplace violence.

Women are at an increased risk of violence, harassment and bullying both in and outside the workplace [9,20]. Due to the dual burden women shoulder both at home and outside, they are exposed to workplace violence in addition to the one they experience at home [9]. The question of women's safety in the workplace is at the nexus of women as workers, as mothers, as victims of violence against women, and sexual and gender-based violence [21]. Ethiopia is at the forefront of the countries where women experience violence of all sorts at home and in the workplace to the extent it become a significant and serious human rights and public health issue [22,23]. Women in Ethiopia are affected by discriminatory policies and gender norms; they are nearly three times more likely be unemployed than men and when they are employed, they are forced to take on vulnerable forms of employment [24]. Above all, differences between women and men's unequal exposure to workplace violence are reinforced by sex-segregated workplaces [16].

According to EU-OSHA [8], the occupational health risks encountered by female workers are especially high for those women who work within the agriculture, the hotels, restaurants

and catering, the transport as well as the manufacturing sectors. In Ethiopia, there is a high level of workplace violence against women that often leads to an extended loss of productive working days [25]. Lack of information about women's rights, limited access to legal services, insensitivity of law enforcement bodies, and the generally poor status of women in the society are some of the main reasons that accounted for the continuation of violence against women in Ethiopia [22].

Findings of previous empirical literatures show exposure of employed women to various types of workplace violence and ill health conditions, including the impacts of such exposure on women's overall physical, biological, psychological, and social life. For instance, a study of women's health working in the mining sector by Botha [26] revealed that women are still exploited and sexually harassed in the mining industry. Moreover, Tawiah, et al. [27] found high level of morbidity in the form of headache, body ache, problems with vision, cough and breathlessness in Brick Kiln and Construction industries of India. Widespread prevalence of sexual harassment among women working in the health care has also been noted [28,29]. Furthermore, the findings of a research in to women's occupational health and safety in the informal economy suggested that maternal market traders face some occupational health risks that have a significant toll on their physical, mental and social health [6]. Moreover, a study of workplace violence among male and female Turkish employees of various sectors undertaken by Akinci, et al. [30] found that nearly half of the workers reported that they had been subjected to a violent event at their workplaces. Workplace violence against women working in the hotel sector, however, is little studied. According to Lippel [7], other types of violence to which women are relatively more exposed than men either at work, because of the nature of their work, or while they travel to and from work, are less often addressed through a gender lens. The characteristics of violence commonly committed against women differ in critical respects from violence commonly committed against men though both can be victims as well as perpetrators of violence [31]. Day to day observations of the hotel, cafeteria, restaurant and liquor selling sectors in Ethiopia reveal that lower level occupational strata and daily routines are predominantly occupied by women while managerial and other "better" positions are held by males. Given the patriarchal nature of the society, women are not only assigned to (often willingly and due to lack of alternatives) less rewarding positions, but also remain highly susceptible to exploitation, gender-based discrimination, and workplace violence. Waitresses working in the sectors under consideration are among these women in Ethiopia whose maltreatment has always been unnoticed and often considered acceptable. Above all, the absence of empirical research findings in the area has also seems to have contributed for the maintenance of the status quo. The objective of the present study was, therefore, to explore workplace violence against women working as waitresses in various hotels, restaurants, cafeterias and grocery stores of Wolaita Sodo town, southern Ethiopia.

## Materials and methods

### Research design

Cross-sectional study was undertaken with descriptive and exploratory study designs. Using qualitative approach, data were collected from a sample of women working as waitresses pertaining to exposure to workplace violence.

### Selection of research participants

The participants of the study were women who have been working as waitresses in different hotels, grocery stores, cafeterias and restaurants of Woliata sodo town, southern Ethiopia. With the aim of analyzing the differential level of exposure to workplace violence within the

group, we purposefully selected equal number of participants representing the group of waitresses working in the restaurant, cafeteria, hotel, and grocery store. Both the recruitment of the research participants and the selection of the specific workplaces were made on the basis of random sampling technique. Because the process of employment was mostly undertaken in the informal networks and the target groups are not beneficiaries of the formal social security schemes of the local government, the total number of waitresses working in the area is unknown. In order to enhance the proportional representation of the sample, we have considered the size of employees working in each category while selecting samples. For instance, compared to others, because large hotels in the area employ relatively higher number of waitresses, the largest proportion of the sample have constituted waitresses from hotels. On the other hand, liquor selling shops and grocery stores were mostly operated by one person in the area and hence, research participants from such category were small. Most important during the sampling process is that we didn't follow as such rigid and predetermined procedures. Instead, the number of waitresses participated in the study was determined on the basis of the point at which data seem to have been saturated. Accordingly, the final sample size participated in the research were 16. Sex and willingness to participate in the study were the major inclusion criteria we used during recruitment. In addition, waitresses who have only recently employed (within in a period of less than a month) were not included in the study.

## Method and procedure of data collection

Qualitative data were collected using in-depth interview method. An interview checklist, containing unstructured questions was prepared to guide the interview.

The questions were prepared in the way they can address the specific research objectives that the research intend to achieve, including: 1) how prevalent is workplace violence against the waitresses and its level of exposure 2) the types of workplace violence to which waitresses were most commonly exposed, 3) the age-sex and other characteristics of the perpetrators that commonly inflict workplace violence, and 4) the common ways of responding/reacting to workplace violence. During field visits, we have first contacted the owners/managers of the randomly selected organizations to get access to the waitresses. Once access was gained, interviewees were informed ahead about the purpose and rationale of undertaking the research and provided their oral consent. Then, they were asked the time and place convenient to them to conduct the interview. Data collectors approached the interviewees in a friendly manner and created a rapport which helped to maintain trust. Research participants were asked unstructured questions, followed by probing in the middle of the interview. Both field notes and voice recordings were used in the process.

## Method of data analysis

Data collected using field notes and voice recordings in the local language were first transcribed and then translated to English. Then, data were organized according to the identified themes. The four major themes identified in the data were 1) exposure to workplace violence, 2) underlying factors contributing to differential exposure to workplace violence within the group, 3) age-sex and related socio-demographic characteristics of perpetrators, and 4) the coping strategies against workplace violence. The data organized under each theme and subtheme were analyzed and interpreted vis-à-vis the specific research objectives already set out at the beginning of the study. therefore, thematic data analysis technique was used in the process of manipulating the data for the particular research purpose at hand.

## Ethical procedures

The researchers obtained ethical approval from the ethical approval committee of Wolaita Sodo University. In addition, a formal letter was secured from the department of sociology, Wolaita Sodo University. An informed written consent was obtained by informing the research participants about the objectives and rationale of undertaking the research. In addition, during reporting of direct quotes as disclosed by the interviewees, we have refrained from citing the exact names of the research participants and used pseudo names where necessary. Moreover, research participants were informed ahead about the possibility of withdrawing from the study in case the need arises at any time. Above all, all methods were carried out in accordance with relevant guidelines and regulations.

## Results

### Victimizations and incidents of workplace violence

Data have shown that waitresses are highly exposed to workplace violence of different forms. The most reported type of workplace violence was verbal abuse, mostly manifested in insisting the victims to provide their phone number to the male customers as an after-service offer. Whereas compliance on the side of the waitress to the telephone number request usually end up with no further violence, failure to positively reply have been reported to cause subsequent forms of verbal and physical violence. In this regard, insulting with derogatory words and humiliation, bulldozing/scaremongering and intimidation are found to be the common ones. For example, an interviewee (P8, 19 years) disclosed: *The most remarkable case I ever experienced was the one I faced in the third month after I joined the work. It happened with a customer who used to come to the restaurant so often and every time he (the customer) comes, he always asks me to give him my phone number. One day, I served him the food he ordered and when he finished eating, he asked me if I could go out with him or give him my phone number. Following my refusal, he warned me that he will not be paying the bill if I continued refusing.* And another research participant (P5, 21 years) added: *I have done as a waitress in many places and I have witnessed so many customers who drink a lot and disturb. Most of them ask me to give them my phone number and to spend the night with them. Sometimes this job is assumed as prostitution.*

In addition to the verbal abuse, incidents of physical violence have also been reported where slapping on the face, beatings, touching on sexual or private parts, spitting, twitting a hand, and other simple physical assaults are commonly experienced by the waitresses. Furthermore, while the victimization of some waitresses was limited to only one or other forms of verbal or physical violence, the case among most interviewees revealed that they are concurrently exposed to both verbal as well as physical violence where the former is followed by the later. The experience of P2 (18 years) would illustrate the scenario:

> *Once upon a time I was working in a grocery store at a small town called Yabelo. On one occasion, two persons came to my workplace, they were too drunk and asked me to spend the night with them but I kept silent. Meanwhile, they started to insult me; but I still kept silent. They then began to touch my buttock and breast. I patiently replied to them that my job is waitress, if they want a woman to sleep with; they can find prostitutes outside. Then they said to me, 'Who are you?' Finally, they started going out without paying the bill; the owner asked me if I took the money. I then replied to him that I didn't and I went to them (customers) to ask. One of them slapped my face and tore my clothes into two pieces; windows of the house were broken during quarrels. Finally, police officers arrived and arrested the man.*

## Dynamics of workplace violence

While we found that waitresses were highly exposed to workplace violence in the study area, research participants also disclosed that the level of exposure to the violence varies across the following circumstances.

**Working in large and small towns.** It was found that incidences of workplace violence against the waitresses tend to increase as one move from larger towns (capital cities) to the smaller ones: . . ...*it varies according to the town where I work. For example, when I worked in Addis Ababa, customers have a lot of respect for the waitresses, but it was very different outside of Addis; there were times when they even think about body posture, customers need a waitress with big thick buttocks. They don't want the service or came here just to eat or drink something. They think they came here to eat good food with "good looking girl". They even prefer between waitresses. They say, 'I don't want you!', even if I am assigned to serve of their sit* (P1, 20 years). Research participants reported to have worked in various towns of Ethiopia hold the view that customers found in the relatively larger towns and regional or capital cities are better educated and hence respectful of the waitresses.

**The situation of the owners/managers.** Data revealed that the incidences of workplace violence against waitresses differ on the basis of the situation of owners/supervisors. While some owners/supervisors attempted to be fair and stand by the side of the waitresses, they commonly preferred to act impartially and help the waitresses during upheavals: *"The supervisor is always by our side. Whenever some customers try to abuse us, she will always tell them that, 'the waitresses are here to serve you, so you shouldn't talk to them like that, and you can leave if you don't want'. As a result, If it is not a customer that comes to our hotel very rarely, such as a b*y *passer, all customers commonly served here do not dare to insult us, they respect us* (P1, 18 years). Conversely, there were supervisors/owners of the organizations that act in favor of the abusive customers. According to the interviewees, such owners or supervisors perceive that 'a customer is a king whatever way he behaves and no matter he abuses the waitresses.'

The narrative of P11 (20 years) stipulates that some owners are not responsive to workplace violence against waitresses: . . ...*when things get worse, I often report to the owner. But, my pain doesn't give him sense; what he need is to maintain the safety of the customers at the expense of ours and keep customers coming to the cafeteria. Though I report what happens with my tears flowing, he never felt my pain. Instead, he gives priority to the customers, not to the waitresses. After all, there is no solution from the owners.* Furthermore, the absence of sympathy among most owners and supervisors is accompanied by the lack of self-esteem and the presence of high job-insecurity of the waitresses all of which have the effect of exacerbating workplace violence against the waitresses. For instance, the respo*nse* of P16 (20 years) that: *It is impossible to judge the owner; after all, it is business for him and I am a waitress here. If I quit, another waitress will come. But, if a customer quits, the owner will lose a lot. Whatever happens, due attention is given to the concerns of the customers, not the waitresses* would illustrate the fact that some waitresses are skeptical of losing their job and willingly opt to accept the abusive conditions.

**Public's insight of the position.** It is found that the understanding to the position of waitress and the value placed to the women assuming the status widely varies across customers. And this is related to the manner in which customers and waitresses interact in the workplace. According to the interviewees, the job of waitress is a less prestigious position and *the assumption held among some customers is that waitresses are part-time prostitutes.* Interviewees believe that one possible source of such perceptions is the dressing norms to which all waitresses must conform which purposively leave some parts of women's body naked in a way it seduces and retains male customers. In addition, it is also disclosed that some customers believe waitresses have the responsibility of satisfying the [sexual] demands of the customers: *Some customers*

*have a distorted view of the job. For example, if you refuse to go out with them, they threaten you saying that they will report to the owner that 'this girl is not comfortable with us' so that the owner fires you away (P3, 21 years).*

**Language.**    The level of exposure to workplace violence has been found to vary on the basis of waitresses' ability to speak the local language. While most of the research participants, especially the ones working in big hotels, disclosed that they came from another area in search of job opportunities, they unveiled that their exposure to workplace violence is partly related to their inability of speaking the local language. It is also found that waitresses face various sorts of workplace violence on accounts of their ethnicity too. For instance, P14 (24 years) said: *Of course, I was attacked not only because of my sex but also because of my ethnicity. When some customers ask me about my ethnicity, I feel discontented. If I disclose to them that I don't speak the language, they tend to question my presence in the area. Some perceive that if I am not able to speak their language, I will not be able to handle them well.* Moreover, the implication of inability to listen and speak the local language on the emotional/psychological wellbeing of waitresses has also been noted in the present study. Data revealed experiences of feelings of loneliness, sense of being marginalized and excluded, and lack of security among waitresses that do not speak the local language.

**The role of the waitresses.**    Data revealed that waitresses often contributed to workplace violence sometimes by initiating discontent among customers. Failure to serve the right order, errors of sequence committed when a customer who came first is not able to be served first, and extended delay both to receive orders and to bring what is ordered have been found to be the ways by which waitresses play the role of initiating violence. According to the research participants, while some customers politely call and complain to the waitress in charge of their sit, others do not give them a chance to correct their mistakes. The following account of clearly depicts the scenario:

*The most memorable day I've ever had was here at XXX hotel; it was a holiday(x-mass) and was a tough day because there were so many people (customers). In the meantime, one of the customers shouted aloud at me for I did not bring the food he ordered me on time. Later, when I brought him the food, he threw it on my face; I was shocked in front of such large gatherings of people. I wish I would have died then. Instead of doing that, it would have been better had he beaten me. I lost control of myself; I was wondering why I'm doing this job*

(P12, 22 years, hotel).

**Customers' behavior and the context.**    The incidence of workplace violence against waitresses has been found to vary depending on customers' behavior during and after drinking alcohol. According to the research participants, workplace violence increases during the night time in which customers get drunk. Week-ends have also been reported to be the days where waitresses are more exposed to the violence since many people; especially employees become free to enjoy and drink alcohol. Furthermore, exposure to workplace violence against women is also influenced by the specific setting in which the waitresses work. We found that waitresses working in grocery stores/bars are relatively more exposed to workplace violence than those working in restaurants and cafeterias.

Data revealed that though waitresses rarely experienced violence from female customers, the most common perpetrators involved in inflicting violence against the victims are males. In addition, differences between male and female customers based on the types of violence they infiltrate to the waitresses have also been noted. In this regard, whereas waitresses mostly become targets of verbal violence from their same-sex customer counterparts, male customers

are reported to engage in all types of violence, including physical violence and sexual abuse. Moreover, we found adults and relatively young customers as the most common perpetrators of violence against the waitresses, while elderlies were reported to be least involved as far as the age of the perpetrators is concerned.

## Coping with workplace violence

Data revealed that waitresses adopt different mechanisms of surviving workplace violence. Ignorance has been reported to be the most commonly practiced type of coping strategy against workplace violence. Nevertheless, when waitresses choose to ignore responding to abusive circumstances, it is done after rational calculation of other alternative way outs. Interviewees disclosed that ignorance is motivated by fear of further consequences: *I will speak nothing, especially to the customer. I might be upset, but I usually choose to immediately leave the area because I know that If I respond countering the deeds of the customer, the worst happens to me to the extent of perhaps losing my job (P1, 19 years)*. The sense of job insecurity observed among most waitresses is the result of the absence of formal procedures before and during employment. Many interviewees disclosed that both their first contact as well as later attachment to their current workplace has taken place through informal networks. Most have reported that they do not have signed contractual agreement specifying the rights and duties of both the employee and the employer and received no letter that clearly stipulates their roles and responsibilities (job description).

The following quote from an interviewee working in a cafeteria demonstrates this reality: "*After completing my 10th grade education, I was just sitting in the house and serving my family because I didn't get a passing grade to continue preparatory education. In the meantime, I asked my friend who used to work in a cafeteria as a waitress to find me a vacant position. She then succeeded in finding it and told me to report to the owner. I was asked to bring a bail. Then, I called my relative who came and signed on a paper to guarantee accountability. It was in this manner that I started working. Except that, there is nothing I singed and received. (*P15,23 years).

While most research participants prefer to pay no attention to quarrelsome circumstances in the workplace, they resort to reporting to the owners/supervisors when ignorance itself couldn't end the process. We also found that waitresses' decision to report to the owner/supervisor heavily relies on their trust that the owner is sympathetic and able to impartially understand their cause: . . .*I was very angry and told the owner everything. Fortunately, the owner and the customer know each other. Then, the owner went and talked to him about what happened. Consequently, the customer paid the bill and finally left. Encountering to such violent customers is a common aspect of our daily life in this work. But, we inform to the boss when the insult becomes worse and she warns them not to come any more if they are about to abuse us like that. She will also tell us not to receive orders from such rude customers anymore* (P9, 20 years).

Contrariwise, waitresses who are aware that the owner/supervisor is not dependable often resort to reporting to the police or opt to shout to the guards of the hotel. Mentioning that there are supervisors/owners that care about the safety of their employees interviewees disclosed that some are less sympathetic that they take for granted violence against waitresses even when it is undertaken in their presence.

In response to facing frequent incidences of violence at a particular workplace and when other previously discussed avenues are proved not helpful, changing a workplace becomes another way some waitresses respond to workplace violence. Interviewees believe that working in relatively bigger hotels is less risky compared to smaller hotels, grocery stores, and cafeterias. A response of P10, 19 years, for instance, demonstrate the views held among waitresses about the less violent nature of customers to be served in the more prestigious hotels: *As a way out, I*

*have now decided not to work in small hotels; I will search for possible opportunities of working in bigger hotels because I think the customers to be served there are well educated and respectful. If I continue facing the same situation there, I change to a new field of work in the future.* Furthermore, we found that some waitresses are studying in the extension modalities at private colleges in the area parallel to working as a waitress. Such waitresses unveiled their plan to quit the present job and their aspiration to be employed with the profession of their college trainings.

Data revealed that research participants vary depending on whether they have been informed ahead and hence, expecting the possibility of workplace violence before joining the sector. It is also found that such variations have influenced the extent to which waitresses differently felt the experience of violence and the way they respond to it. Whereas some interviewees disclosed that they were conscious of and expected the likelihood of facing workplace violence, they tend to assume the violence as the normal aspects of the work and are relatively less aggressive in reacting to the incidences. For instance, P12, 22 years disclosed: *obviously, I had the awareness; it was my cousin who suggested me to do this job. Because she was a waitress I used to hear from her when she was talking about the heavy workload and her ample experiences of diverse workplace violence. I finally found out that what I have faced and still facing is the same to her. But I already convinced myself before beginning the job. Even though I became angry, cried a lot, and felt lonely but I do not give up because I hope tomorrow will be fine.* Others have reported experiences of shocks while encountering the situation being under circumstances of lacking prior knowledge. For instance, the story of P6, 18 years demonstrates that the shock is temporary and some of such waitresses would prefer to remain in the work withstanding the shocks: *I had no idea about the job before joining it. I came into it all of a sudden following my failure to get a passing mark in 10<sup>th</sup> grade. I was unemployed for a long time, and didn't think I would do this job; I thought I would start my own business. Subsequently, I joined the work just considering it as a temporary solution. Even if I frequently face different maltreatments, I try to convince myself that this time will pass and good days will come and persuaded myself to be patient.*

## Discussion

The present study particularly focused on women working as waitresses in the various hotels, restaurants, grocery stores, and cafeterias of Wolaita Sodo town, southern Ethiopia. Qualitative data were collected by interviewing 16 (sixteen) waitresses and thematically analyzed. Accordingly, the results of the study have shown that the incidence of workplace violence against waitresses in the study area is high. This result is consistent to Yemane [22] who found that Ethiopia has one of the highest prevalence of sexual, emotional, and physical violence against women in the work place. According to Abera, et al. [25], there is a high level of workplace injuries that often leads to an extended loss of productive working days in Ethiopia, where occupational safety and health services are inadequately organized. Moreover, a study of workplace violence among Turkish employees of various sectors undertaken by Akinci, et al. [30] found that nearly half of the workers reported that they had been subjected to a violent event at their workplaces. According to CCOHS [19] working with customers or the public is among the work factors that put people at increased risk of workplace violence and sellers of alcohol are among the occupational groups that tend to be at risk from workplace violence. Women are at particular risk of violence, harassment and bullying both in and outside the workplace [9,20].

The most commonly reported type of workplace violence in the present study is verbal abuse, mostly manifested in insisting the victims to provide their phone number to the male

customers. Furthermore, waitresses are also found to be exposed to other forms of verbal abuses including insulting and humiliation, bulldozing/scaremongering and intimidation. In addition to the verbal abuse, incidents of physical violence have also been reported where slapping on the face, beatings, touching on sexual or private parts, spitting, twitting a hand, and simple physical assaults are commonly experienced by the waitresses. The finding is consistent to Akinci, et al. [30] who found that the most common type of violence was verbal abuse, followed by bullying, physical assault and sexual harassment. A study of women's health working in the mining sector by Botha [26] revealed that women are still exploited and sexually harassed in the mining industry. Incidents taking place daily vary from whistling; name calling; use of vulgar or derogatory language; display of body parts; physical contact, ranging from touching to sexual assault and rape; to the exchange of sexual favors for promotion. Melak [32] found a high prevalence of violence against female university students the rate of which ranges from the most frequent form, i.e., verbal harassment in the form of insult and catcall, to complete or attempted rapes which are the rare type of sexual violence against female students.

While we found high incidences of workplace violence against women in the study area, research participants also disclosed that the level of exposure to the violence varies across various circumstances, including working in large and small towns, the situation of the owners/supervisors, public's insight of the position, waitresses' ability to speak the local language, the role of the waitresses, customers' behavior and the specific context in which waitresses work. Workplace violence against women is rooted in perpetrator's sense of power, control, and entitlement, gender-based stereotypes, perpetrator's use of alcohol and other drugs, and victim's blaming ideas, such as the belief that a victim somehow "asked for it" by the way s/he behaves, dresses or lives [33]. According to ILO [16], risk factors for victims include their age, experience, being female, and attitudes and expectations (perceived vulnerability). Workplace risk factors are comprised of the working environment such as organizational setting and managerial style, as well as the workplace culture and external environment. Task-related risk factors include working alone, with the public and/or people in distress, and in educational settings. The main causes for the exposure of females to different forms of violence are legal and structural, such as the absence of sound policy on girls in general and gender-based violence in particular. In addition, the absence of gender sensitivity in the legislations of organizations and the absence of a clear mandate and authority for the existing gender focal point institution is also another serious challenge [32]. The existence of relevant national legislative frameworks on gender-based violence as well as a workplace-level policy and implementation strategy are also important contextual factors [16]. Concerns of workers' occupational health and safety have been given attention in Ethiopian Federal Negarith Gazeth proclamation no. 1/1995 [34]. Article 42 (2) of the proclamation, for instance, provides that workers have the right to reasonable limitation of working hours, to rest, leisure, to periodic leaves with pay, to remuneration for public holidays as well as healthy and safe work environment.

Findings of the present study reveal that waitresses adopt different mechanisms of surviving workplace violence. Ignorance has been reported to be the most commonly practiced type of coping strategy against workplace violence, followed by reporting to the owners/supervisors, and changing a workplace. The result is once again consistent to Akinci, et al. [30] who found that victims preferred to ignore or deny the violent act instead of fighting against it. According to their study, the most frequently reported way victims react to workplace violence was talking to colleagues about the violent act, followed by talking with family and friends and warning the perpetrator, "doing nothing," "pretending nothing had happened", and "getting help from the police." Fredrickson [35] found that the main reasons that employees do not report workplace violence include fear of their supervisor's reaction, lack of company procedures/policies, lack of training, becoming the 'office snitch', and fear of retaliation. According to the finding

of the Fredrickson's study, employees afraid to report potential workplace violence to their supervisor because they think the supervisor will over-react or think they are unable to handle their own job/responsibilities. Lack of information about women's rights, limited access to legal services, insensitivity of law enforcement bodies, and the generally poor status of women in the society are some of the main reasons that accounted for the continuation of violence against women [22]. Above all, workplace violence can be considered as a reflection of the general and increasing phenomenon of violence in many areas of social life which has to be broadly viewed at the level of the whole society [12]. The findings of the present study significantly contribute to existing knowledge on workplace violence and gender-based violence. Above all, it narrows the gap in the empirical literature as far as the workplace violence faced by women working in hotels, restaurants, cafeterias, and grocery stores is concerned.

## Conclusion

The results of the research indicated that waitresses are highly exposed to workplace violence both on account of their occupation of presumably a 'less prestigious position' and their sex. While waitresses are generally exposed to all forms of violence including spitting, throwing objects, simple physical assault, touching on private parts, and intimidation, verbal abuse and emotional exploitations are found to be the most experienced ones. Although waitresses rarely experience violence from female customers, the most common perpetrators involved in the violence were males. Above all, whereas the vulnerability of waitresses to workplace violence has been noted, level of exposure to the violence varies depending on the personal qualities of the owners/supervisors, the nature of the workplace itself and the socio-economic status of the customers served in those workplaces, waitresses' ability to speak the local language, peoples' perception of the position and responsibility of the waitresses, the manner in which the waitresses serve the customers, and the particular context in which waitresses work. Ignorance, mostly motivated by fear of losing one's job, was the common way waitresses respond to the violence. The study implies the vulnerability state of waitresses partly due to lack of adequate awareness regarding the rights and obligations expected of an employee when working in such organizations. This is again exacerbated by the lack of formal employment procedures in such organizations. These circumstances have contributed to waitresses' lack of job security and the fear of other possible further consequences when deciding to react to violence in the way it safeguards their rights. Furthermore, the wrong perception of the public about the position of waitress and the absence of concern from the side of the owners about workplace violence are also other aggravating factors. Hence, awareness creation and supervisory activities are expected from the media, workers' and social security office of the government, police and other concerned bodies.

## Supporting information

**S1 File. In-depth interview guide 1.**
(DOCX)

## Acknowledgments

First of all, we would like to forward our heartfelt gratitude to the owners and managers of the hotels, cafeterias, restaurants and grocery stores in Wolaita Sodo town who have positively replied to our request and allowed the initial contact with the waitresses. Moreover, all the participants of the study who have devoted their time in providing genuine data also deserve appreciation.

## Author Contributions

**Conceptualization:** Bewunetu Zewude.

**Data curation:** Tewodros Habtegiorgis.

**Formal analysis:** Bewunetu Zewude.

**Investigation:** Bewunetu Zewude.

**Methodology:** Bewunetu Zewude, Tewodros Habtegiorgis.

**Supervision:** Bewunetu Zewude.

**Validation:** Bewunetu Zewude.

**Writing – original draft:** Bewunetu Zewude, Tewodros Habtegiorgis.

**Writing – review & editing:** Bewunetu Zewude.

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
