## [Decision Letter · Decision Letter 0]

24 Jun 2021

PONE-D-21-09783

Victimizations and Surviving of Workplace Violence against Waitresses in Southern Ethiopia

PLOS ONE

Dear Dr. Zewude,

Thank you for submitting your manuscript to PLOS ONE. After careful consideration, we feel that it has merit but does not fully meet PLOS ONE’s publication criteria as it currently stands. Therefore, we invite you to submit a revised version of the manuscript that addresses the points raised during the review process.

As the reviewers mentioned, the findings from this paper have the potential to make important contributions to the work-pace violence field in lower- and middle-income countries, specifically Ethiopia. There are, however, some important issues that need to be addressed for this paper to move forward in the review process. In particular, there needs to be additional clarification/detail about the sampling for the study. Authors need to provide information about the ethical process and considerations in the study (including consent process, protection of participants, confidentiality, and ethics review and approvals). The authors should consider the use of pseudonyms or participant ID codes and inclusion of a statement or two in the methods section clarifying the use of pseudonyms/IDs and/or demographic details of participants (e.g., age). Please also clarify/highlight/discuss key findings and provide a discussion of implications of findings for policy/practice/intervention/future research in the discussion/conclusion.

We look forward to receiving your revised manuscript.

Kind regards,

Samantha C Winter, Ph.D.

Academic Editor

PLOS ONE

Journal Requirements:

2. Please include a copy of the interview guide used in the study, in both the original language and English, as Supporting Information, or include a citation if it has been published previously.

Reviewers' comments:

Reviewer's Responses to Questions

**Comments to the Author**

1. Is the manuscript technically sound, and do the data support the conclusions?

Reviewer #1: Yes

Reviewer #2: Yes

2. Has the statistical analysis been performed appropriately and rigorously? 

Reviewer #1: N/A

Reviewer #2: N/A

3. Have the authors made all data underlying the findings in their manuscript fully available?

Reviewer #1: Yes

Reviewer #2: Yes

4. Is the manuscript presented in an intelligible fashion and written in standard English?

Reviewer #1: Yes

Reviewer #2: Yes

5. Review Comments to the Author

Reviewer #1: First, I would like to thank the author/researcher in bringing out the issues of workplace violence among female in a developing country. This article will add up some more information in relevant topic. However, it needs more effort in revision. I think the manuscript should be independent in terms of its format rather than being a short-cut version of a research report/thesis. Below are the general comments (except for the specific comments made throughout the manuscript by doing track changes – see attachment) that needs main revision.

Introduction – This section looks good, but it has plenty of information. It needs to be shortened by condensing the information that are related to workplace violence and women, as well as by removing the duplication (in information) and other theoretical description that are not specific with the objectives of the study. For example, this paper only focuses on the status of workplace violence and does not have anything related to policy/legislative aspects. So, it is better to remove that.

Materials and methods – Overall good. However, it needs a bit more clarification on the selection of participants (e.g., the number of participants selected from each store). The method of data analysis needs more detailed description on how the data were analyzed. Ethics related aspects (privacy, confidentiality, ethical approval, etc.) are missing.

Results – This section is very good except the style of presenting the information (layout) and some grammatical errors. Structure or layout needs to be rearranged, such as change of paragraph while mentioning the statement of participants. Some statements given by participants are in present tense, which needs to be corrected. There are names in few statements which does not seems to be ethical. Participant's name must be removed and de-identified, but it does need the other characteristics of participants such as their age and working store. E.g. P12, 26 years, Cafeteria

Discussion – The starting of the discussion section does not seem to be appropriate. It should be started with the background of the results/objectives. Do not repeat the whole results but briefly mention your results and discuss it with other. While discussing, focus on why and how of your findings as well as other study's findings. At the end of discussion section, write few sentences on how your findings are significant or different than other studies and its implication in Ethiopian context.

References – Referencing in the text as well as in the bibliography section does not match the PLOS guidelines. It is meant to be in Vancouver style, but this manuscript adopted Author-Date/Harvard method. It needs major revision. Moreover, references need to be cross checked between the text and the bibliography section.

Reviewer #2: 1. It would be good to have a theory for the study

2. Methodology:

a. The sampling framework needs to be clear. For example, Woliata sodo sounds like a small town and it would be important to mention about how many groceries, restaurants etc that are there, so that we can have a feel of the representation.

b. “We have purposefully selected equal number of participants representing the group of waitresses working in the restaurant, cafeteria, hotel, and grocery store”- this statement can be challenged because some of the places may have fewer workers than others, which means that some organization may end up having more participants than others. Could there be some restaurants, cafeteria having fewer women employees than the selected ten?

c. Another question is the criteria used to select the ten from each organization?

d. Whereas it is fair to say that “data saturation was attained after sixteen waitresses were interviewed”; one is left wondering about the representation of the 16---was there a good representation from the four sectors, that is, restaurant, cafeteria, hotel, and grocery?

e. “Sex and willingness to participate in the study were the major inclusion criteria we used”; again, the question is whether each person was approached and were asked whether they were willing to participate?

f. Ethical issues have not been addressed in this very sensitive topic

g. Research approvals have not been mentioned

3. Results & Discussions: they are very interesting, but it would have been interesting to hear voices from the different industries. For example, from the analysis, what kind of violence is mainly experienced by those in the cafeteria industry/ or to describe the direct quote verses (e.g. said so and so who works in the grocery store. Further, the language needs to be reworked on in this section. For example, is the coping strategy ignorance or ignoring – the two terms are different, and it feels like they have been used interchangeably. Direct quotes need to be restructured so that they make sense to the readers. Some of the paragraphs are too long especially under the discussion heading.

4. In the discussion section, the key findings need to be captured as the existing literature/ empirical findings are discussed.

5. Make the conclusion strong so that we hear the gap that existed and how it has been met through this study.

6. PLOS authors have the option to publish the peer review history of their article (what does this mean?). If published, this will include your full peer review and any attached files.

Reviewer #1: No

Reviewer #2: No

---

## [Author Response · Author response to Decision Letter 0]

25 Jun 2021

Point-by-point Response

Dear Dr. Samantha, we would like to express our sincere gratitude to you for your editorial services and your help in the process of enabling the peer review process to be undertaken within a reasonable time frame. We are also thankful to the reviewers that have devoted their time for reviewing the material and contributed their share for improving the quality of the article. We wholeheartedly believe that the comments and suggestions provided herewith are highly helpful in terms of producing an interesting paper. 

Dear all, below is a table containing the comments of the reviewers (and the editor) and a point-by-point response of the authors. We believe that you will find satisfying responses for all the concerns you raised in the paper. If there are further issues that need to be addressed, please do not hesitate to contact us any time. Due to our ambition to get the article published very soon, the corresponding author will be staying 7/24 online attending your emails. 

Dear editor, in an attempt to ease the editorial process and thereby fasten the publication, we have revised the reference section and changed the style of the in-text citation (in superscript form) according to the journal’s style. Above all, the data collection instrument has also been attached at the end of this page. 

Thank you very much once again!

source of Comment Details of Comments Authors’ Responses 

 Reviewer #1 • Introduction –-This section looks good, but it has plenty of information. It needs to be shortened by condensing the information that are related to workplace violence and women, as well as by removing the duplication (in information) and other theoretical description that are not specific with the objectives of the study. For example, this paper only focuses on the status of workplace violence and does not have anything related to policy/legislative aspects. So, it is better to remove that. • Corrected accordingly! 

• In addition to correcting all the recommended points, we have also revised the whole introductory section, to the extent of paraphrasing it. 

 • Materials and methods –Overall good. However, it needs a bit more clarification on the selection of participants (e.g., the number of participants selected from each store). The method of data analysis needs more detailed description on how the data were analyzed. Ethics related aspects (privacy, confidentiality, ethical approval, etc.) are missing. • Corrected accordingly!

• The ethics statement is provided as a separate section, data analysis method clarified, sampling again clarified.

 • Results –This section is very good except the style of presenting the information (layout) and some grammatical errors. Structure or layout needs to be rearranged, such as change of paragraph while mentioning the statement of participants. Some statements given by participants are in present tense, which needs to be corrected. There are names in few statements which does not seems to be ethical. Participant's name must be removed and de-identified, but it does need the other characteristics of participants such as their age and working store. E.g. P12, 26 years, Cafeteria • corrected accordingly! 

 • The starting of the discussion section does not seem to be appropriate. It should be started with the background of the results/objectives. Do not repeat the whole results but briefly mention your results and discuss it with other. While discussing, focus on why and how of your findings as well as other study's findings. At the end of discussion section, write few sentences on how your findings are significant or different than other studies and its implication in Ethiopian context. • Corrected according to the suggestion. 

• As far as inclusion of just major findings is concerned in the discussion, we have made at most effort of not missing interesting findings while at the same trying not to go to unnecessary details. 

 • Referencing in the text as well as in the bibliography section does not match the PLOS guidelines. It is meant to be in Vancouver style, but this manuscript adopted Author-Date/Harvard method. It needs major revision. Moreover, references need to be cross checked between the text and the bibliography section. • corrected accordingly 

Reviewer #2 • It would be good to have a theory for the study • Dear reviewer, thank you for your insightful suggestion. we are afraid that inclusion of additional literatures would make the introductory section unnecessarily too much. Of course, it could have been done by reducing some portions from the introductory section. But we still believe that sufficient background and justification have been provided in the section. We hope you will understand and share our concern.

 • The sampling framework needs to be clear. For example, Woliata sodo sounds like a small town and it would be important to mention about how many groceries, restaurants etc that are there, so that we can have a feel of the representation. • Dear reviewer, it is true that Wolaiata Sodo is relatively a small town-actually not that too small. The problem is not from the size of the town, rather it is about the availability of data from the concerned gov’t office. When asked, what the officers give us is just an estimation arrived without data. 

• In fact, we have made certain clarifications to the sampling section

 • “We have purposefully selected equal number of participants representing the group of waitresses working in the restaurant, cafeteria, hotel, and grocery store”- this statement can be challenged because some of the places may have fewer workers than others, which means that some organization may end up having more participants than others. Could there be some restaurants, cafeteria having fewer women employees than the selected ten? • we have revised the section accordingly!

• thank you

 • Another question is the criteria used to select the ten from each organization? • revised 

 • Whereas it is fair to say that “data saturation was attained after sixteen waitresses were interviewed”; one is left wondering about the representation of the 16---was there a good representation from the four sectors, that is, restaurant, cafeteria, hotel, and grocery? • the section has been revised accordingly

 • “Sex and willingness to participate in the study were the major inclusion criteria we used”; again, the question is whether each person was approached and were asked whether they were willing to participate? • Willingness of waitresses to participate in the study was asked after inclusion at the initial sampling stage-40. Dear reviewer, the 16 were the ones who finally given their full consent to participate. 

 • Ethical issues have not been addressed in this very sensitive topic • now addressed 

 • Research approvals have not been mentioned • now considered 

 • Results & Discussions: they are very interesting, but it would have been interesting to hear voices from the different industries. For example, from the analysis, what kind of violence is mainly experienced by those in the cafeteria industry/ or to describe the direct quote verses (e.g. said so and so who works in the grocery store. Further, the language needs to be reworked on in this section. For example, is the coping strategy ignorance or ignoring – the two terms are different, and it feels like they have been used interchangeably. Direct quotes need to be restructured so that they make sense to the readers. Some of the paragraphs are too long especially under the discussion heading. • revised 

 • In the discussion section, the key findings need to be captured as the existing literature/ empirical findings are discussed. • revised 

 • Make the conclusion strong so that we hear the gap that existed and how it has been met through this study. • Considered. The gaps and how they were addressed, the difference with previous studies, and the contribution of the study all are discussed in the discussion in addition to the conclusion. 

In-depth Interview guiding questions

1. How prevalent is gender-based violence in your work place?

1.1. Have you ever faced gender-based violence now or before while working in similar work places? 

1.2. Would you please tell me some of the violence types that you faced?

1.3. ………..also ask her to specify where, when (what time), by whom ( whether she has faced it from the side of customer/client, employer/owner, owner’s family/relative, stranger, etc)

1.4. ……also specify if she faced the violence on account of her gender, ethnicity, religion, social class, or any other stratifying variable?

1.5. Have you ever faced violence or discrimination on the basis of your ethnicity, religious affiliation, or belongingness to a social group that is perceived to as a different/minority in the area?

2. How did you react or respond to the violence you have faced?

2.1. What did you do after facing the violence or discrimination in your work place?

2.2. Have you ever reported to the police, managers/supervisors, [body] guards, the owner, etc……or you just kept silent?

2.3. ……….and why did you do that ……or not did so?

2.4. If you have ever reported, then how did they/it respond to your complaints?

2.5. …………..What is the most common thing you do whenever you face such violence?

3. What physical, psychological, and social challenges have you faced because of the violence? 

……… social exclusion, physical damage, illness or sickness, fear or feelings of intimidation, depression, anxiety, loneliness, feelings of helplessness, feelings of worthlessness, etc

4. What was your expectation regarding the possibility of being exposed to gender-based violence in such work places? 

4.1. Have you ever expected or were you expecting to be exposed for such workplace violence when you decide to work here?

4.2. ….if yes, then, have you convinced yourself to accept victimizations?

4.3. …….what strategies have you planned to cope-up with the situation?

No (code) Variable Categories Frequency 

 Age 

 Marital status 

 Education 

 Years in the work

---

## [Decision Letter · Decision Letter 1]

20 Sep 2021

PONE-D-21-09783R1Victimizations and Surviving of Workplace Violence against Waitresses in Southern EthiopiaPLOS ONE

Dear Dr. Zewude,

Thank you for submitting your manuscript to PLOS ONE. After careful consideration, we feel that it has merit but does not fully meet PLOS ONE’s publication criteria as it currently stands. Therefore, we invite you to submit a revised version of the manuscript that addresses the points raised during the review process.

The authors made important changes to the manuscript in the revision; however, the manuscript needs some additional revision before acceptance for publication. The authors changed the in-text referencing style from one which used author names and dates such as APA (e.g., Adams, 2019) to a numerical referencing style; however, when they did this, all of the references to specific studies in the introduction and throughout the discussion became inappropriately referenced, (e.g. according to [8],...). These need to be corrected. Additional proofreading for typographical, editorial, and grammatical errors is needed. Please see detailed reviewer comments below.

We look forward to receiving your revised manuscript.

Kind regards,

Samantha C Winter, Ph.D.

Academic Editor

PLOS ONE

Journal Requirements:

Additional Editor Comments (if provided):

Reviewers' comments:

Reviewer's Responses to Questions

**Comments to the Author**

1. If the authors have adequately addressed your comments raised in a previous round of review and you feel that this manuscript is now acceptable for publication, you may indicate that here to bypass the “Comments to the Author” section, enter your conflict of interest statement in the “Confidential to Editor” section, and submit your "Accept" recommendation.

Reviewer #1: (No Response)

Reviewer #2: (No Response)

2. Is the manuscript technically sound, and do the data support the conclusions?

Reviewer #1: Yes

Reviewer #2: Yes

3. Has the statistical analysis been performed appropriately and rigorously? 

Reviewer #1: N/A

Reviewer #2: Yes

4. Have the authors made all data underlying the findings in their manuscript fully available?

Reviewer #1: Yes

Reviewer #2: Yes

5. Is the manuscript presented in an intelligible fashion and written in standard English?

Reviewer #1: Yes

Reviewer #2: Yes

6. Review Comments to the Author

Reviewer #1: I would like to thank authors for their effort in revising the manuscript. Although most of the comments have been addressed, the manuscript still needs revision especially in sentence structure, grammar, and format/style. I highly recommend authors to take referencing pattern/citation seriously and revise it. Also, I recommend the authors to seek professional editing for grammar and sentence structure.

I have highlighted some errors in the file but those are not all. Please do a thorough proof reading.

Below are some of the errors that still exist in the manuscript.

In abstract (very initial pages, not the text one), correct the capitalization on keywords.

There are still some issues in referencing. For example, in the introduction section, there are few sentences that says According to [8], various sectors undertaken by [30], health working in the mining sector by [26] revealed. This is not the way how you do citation. You can write the author's name in this type of sentences (although it is in Vancouver) and mention the references in number at the end of the sentence OR write it as a finding and keep the reference at the end. For example ----- Instead of writing "Moreover, [27] found high level of morbidity in the form of headache, body ache, problems with vision, cough and breathlessness in Brick Kiln and Construction industries of India" simply write "high level of morbidity in the form of headache, body ache, problems with vision, cough and breathlessness in Brick Kiln and Construction industries of India [27].

No need to put reference in the first sentence of 'research design'

Keep an eye on the capitalization of middle words. Example – "Selection of Research participants"

Under the section "selection of research participants", there are notable errors in grammar. Methodology should be always in past tense, not in present tense. Same types of errors still exist in 'Findings'. Example – "Data reveal that waitresses adopt", "data reveal that waitresses often contribute to", "Data revealed that though waitresses rarely experience"…… and so on.

In findings section, it's good to break the paragraph rather than placing chunk of information and quotations in one paragraph. For example, see the sub-section "the situation of the owners/manager".

To sum up, revision stll requires in sentence structure, grammar, and references.

Reviewer #2: The author(s) have improved on the document and critically looked into the issues that were raised during the first review. The issues has been presented well and I hope the findings will get to the necessary policy makers or influencers of the same.

7. PLOS authors have the option to publish the peer review history of their article (what does this mean?). If published, this will include your full peer review and any attached files.

Reviewer #1: No

Reviewer #2: **Yes: **Dr. Julia Kagunda

---

## [Author Response · Author response to Decision Letter 1]

21 Sep 2021

source of Comment Details of Comments Authors’ Responses 

 Reviewer #1 • Although most of the comments have been addressed, the manuscript still needs revision especially in sentence structure, grammar, and format/style. I highly recommend authors to take referencing pattern/citation seriously and revise it. • Revised 

 • In abstract (very initial pages, not the text one), correct the capitalization on keywords. • Corrected 

 • There are still some issues in referencing. For example, in the introduction section, there are few sentences that says According to [8], various sectors undertaken by [30], health working in the mining sector by [26] revealed. This is not the way how you do citation. You can write the author's name in this type of sentences (although it is in Vancouver) and mention the references in number at the end of the sentence OR write it as a finding and keep the reference at the end. For example ----- Instead of writing "Moreover, [27] found high level of morbidity in the form of headache, body ache, problems with vision, cough and breathlessness in Brick Kiln and Construction industries of India" simply write "high level of morbidity in the form of headache, body ache, problems with vision, cough and breathlessness in Brick Kiln and Construction industries of India [27]. • Revised 

 • No need to put reference in the first sentence of 'research design' • Removed 

 • Keep an eye on the capitalization of middle words. Example – "Selection of Research participants"

Under the section "selection of research participants", there are notable errors in grammar. Methodology should be always in past tense, not in present tense. Same types of errors still exist in 'Findings'. Example – "Data reveal that waitresses adopt", "data reveal that waitresses often contribute to", "Data revealed that though waitresses rarely experience"…… and so on. • Revised 

 • In findings section, it's good to break the paragraph rather than placing chunk of information and quotations in one paragraph. For example, see the sub-section "the situation of the owners/manager". • Corrected 

Reviewer #2 • The author(s) have improved on the document and critically looked into the issues that were raised during the first review. The issues has been presented well and I hope the findings will get to the necessary policy makers or influencers of the same. • Thank you

Editors’ comments • The authors changed the in-text referencing style from one which used author names and dates such as APA (e.g., Adams, 2019) to a numerical referencing style; however, when they did this, all of the references to specific studies in the introduction and throughout the discussion became inappropriately referenced, (e.g. according to [8],...). These need to be corrected. • Revised 

 • Additional proofreading for typographical, editorial, and grammatical errors is needed. • Revised

---

## [Decision Letter · Decision Letter 2]

14 Dec 2021

Victimizations and Surviving of Workplace Violence against Waitresses in Southern Ethiopia

PONE-D-21-09783R2

Dear Dr. Zewude,

We’re pleased to inform you that your manuscript has been judged scientifically suitable for publication and will be formally accepted for publication once it meets all outstanding technical requirements. Please take a look at Reviewer #1's comments about reviewing the manuscript for technical and grammatical issues, and correcting the statements they pointed out in their final comments (attached below).

Kind regards,

Samantha C Winter, Ph.D.

Academic Editor

PLOS ONE

Additional Editor Comments (optional):

Reviewers' comments:

Reviewer's Responses to Questions

**Comments to the Author**

1. If the authors have adequately addressed your comments raised in a previous round of review and you feel that this manuscript is now acceptable for publication, you may indicate that here to bypass the “Comments to the Author” section, enter your conflict of interest statement in the “Confidential to Editor” section, and submit your "Accept" recommendation.

Reviewer #1: All comments have been addressed

Reviewer #2: All comments have been addressed

2. Is the manuscript technically sound, and do the data support the conclusions?

Reviewer #1: Yes

Reviewer #2: Yes

3. Has the statistical analysis been performed appropriately and rigorously? 

Reviewer #1: N/A

Reviewer #2: N/A

4. Have the authors made all data underlying the findings in their manuscript fully available?

Reviewer #1: No

Reviewer #2: Yes

5. Is the manuscript presented in an intelligible fashion and written in standard English?

Reviewer #1: Yes

Reviewer #2: Yes

6. Review Comments to the Author

Reviewer #1: It has lots of improvements now. The only thing that needs further improvement (in few places) is the layout based on the journal requirements (e.g. see the subsections of 'results'). I hope this would be corrected during the editing process. Please edit the grammar for the following sentences from 'results' section which needs to be in the past tense.

The narrative of P11 (20 years) stipulates that some owners are not responsive to workplace violence against waitresses: (owners were not)

Data revealed that waitresses adopt different mechanisms of surviving workplace violence. (reveals and adopted)

While most research participants prefer to pay no attention to quarrelsome circumstances in the workplace, they resort to reporting to the owners/supervisors when ignorance itself couldn’t end the process. (preferred to pay)

Contrariwise, waitresses who are aware that the owner/supervisor is not dependable often resort to reporting to the police or opt to shout to the guards of the hotel. (who were aware)

Although waitresses rarely experience violence from female customers, the most common perpetrators involved in the

violence were males (In Conclusion) (rarely experienced)

It is good to go. However, I request the authors to re-read the manuscript because all the grammatical errors (minor) could not be reflected in reviewer's comment and those can corrected by the authors themselves.

Thank you.

Reviewer #2: The review comments by the reviewer have been worked on.

The only outstanding thing is the paragraphs are very long and need to be worked on. The following have been worked on:

a. Methodology issues like the selection according to the ‘industry- hotel, restaurant, grocery and store has been explained and the representation of the same;

2. Results & Discussions: Different types of violence discussed and the most severe highlighted while the discussion is rich, including other empirical studies

3. Make the conclusion strong so that we hear the gap that existed and how it has been met through this study.

7. PLOS authors have the option to publish the peer review history of their article (what does this mean?). If published, this will include your full peer review and any attached files.

Reviewer #1: No

Reviewer #2: No

---

## [Editor Report · Acceptance letter]

16 Dec 2021

PONE-D-21-09783R2 

Victimizations and Surviving of Workplace Violence against Waitresses in Southern Ethiopia 

Dear Dr. Zewude:

I'm pleased to inform you that your manuscript has been deemed suitable for publication in PLOS ONE. Congratulations! Your manuscript is now with our production department. 

Kind regards, 

on behalf of

Dr. Samantha C Winter 

Academic Editor

PLOS ONE